# Organozinc pivalates for cobalt-catalyzed difluoroalkylarylation of alkenes

Xinyi Cheng[1,3], Xingchen Liu[1,3], Shengchun Wang[2,3], Ying Hu[1], Binjing Hu[1], Aiwen Lei 🄳 [2✉] & Jie Li[1✉]

Installation of fluorine into pharmaceutically relevant molecules plays a vital role in their properties of biology or medicinal chemistry. Direct difunctionalization of alkenes and 1,3-dienes to achieve fluorinated compounds through transition-metal catalysis is challenging, due to the facile β-H elimination from the $Csp^3$–[M] intermediate. Here we report a cobalt-catalyzed regioselective difluoroalkylarylation of both activated and unactivated alkenes with solid arylzinc pivalates and difluoroalkyl bromides through a cascade $Csp^3$–$Csp^3$/$Csp^3$–$Csp^2$ bond formation under mild reaction conditions. Indeed, a wide range of functional groups on difluoroalkyl bromides, olefins, 1,3-dienes as well as (hetero)arylzinc pivalates are well tolerated by the cobalt-catalyst, thus furnishing three-component coupling products in good yields and with high regio- and diastereoselectivity. Kinetic experiments comparing arylzinc pivalates and conventional arylzinc halides highlight the unique reactivity of these organozinc pivalates. Mechanistic studies strongly support that the reaction involves direct halogen atom abstraction via single electron transfer to difluoroalkyl bromides from the in situ formed cobalt(I) species, thus realizing a Co(I)/Co(II)/Co(III) catalytic cycle.

[1] Key Laboratory of Organic Synthesis of Jiangsu Province, College of Chemistry, Chemical Engineering and Materials Science, Soochow University, Suzhou, People's Republic of China. [2] College of Chemistry and Molecular Sciences, the Institute for Advanced Studies (IAS), Wuhan University, Wuhan, People's Republic of China. [3]These authors contributed equally: Xinyi Cheng, Xingchen Liu, Shengchun Wang. ✉email: aiwenlei@whu.edu.cn; jjackli@suda.edu.cn

Organometallic reagents, due to their versatile reactivity and high functional group compatibility, have been broadly used for the synthesis of pharmaceuticals and agrochemicals. Especially, transition metal-catalyzed cross-coupling strategy, is of great importance for the development of modern organic chemistry[1–6]. Among them, boron[7–10] and zinc organometallics[11–15] have found extensive applications in such coupling reactions for the preparation of a wide range of highly complex molecules. As compared with the bench-stable, commercially available boronic derivatives, conventional organozinc reagents (RZnX, X = Cl, Br, I) displayed even more exquisite reactivity under mild reaction conditions in many coupling reactions; however, highly air- and moisture-sensitive still represent drawbacks for their synthetic applications. Remarkably, Knochel and coworkers recently developed a solid organozinc pivalate (RZnX·Mg(OPiv)$_2$·LiCl, which is abbreviated henceforth as RZnOPiv for the sake of clarity)[16], which shows greatly enhanced air and moisture stability after solvent evaporation[17]. These organozinc reagents, including (hetero)aryl[17–22], alkynyl[23,24], and alkyl zinc pivalates[25], exhibited good reactivity in Pd-catalyzed cross-coupling reactions with unsaturated halides, which further proved to be broadly applicable for the late-stage functionalizations of biologically active molecules[26]. Although palladium catalysts are very useful, the 3d transition metals[27–34], especially cobalt[35–39], have recently found numerous applications due to its low toxicity, low cost, and natural abundance. Recently, (hetero)arylzinc pivalates have been successfully used for cobalt-catalyzed cross-couplings with unsaturated halides[40], alkenyl acetates[41], N-hydroxylamine benzoates[42], anthranils[43], N-hydroxyphthalimide[44], as well as thiopyridyl ester derivatives[45]. Therefore, the ease of preparation, stability, and exquisite reactivity of these solid zinc reagents have attracted considerable attention of synthetic chemists.

Transition-metal-catalyzed regioselective difunctionalizations of olefins with two different functional groups have been recognized as an increasingly viable tool for preparing complex organic compounds from readily available starting materials[46–48]. However, due to the facile β–H elimination from the $Csp^3$-[M] intermediate[49–52], it still remained challenging to construct two C–C bonds through transition-metal catalyzed multicomponent dicarbofunctionalization of alkenes (Fig. 1a)[53–55]. Importantly, highly regioselective Ni-catalyzed alkylarylation of vinylarenes with alkyl halides and arylzinc iodides has been recently developed by Giri and coworkers[56], they further extended the substrate scope to α-halocarbonyl derivatives (Fig. 1a)[57]. Besides, the installation of fluorine into bioactive molecules uniquely plays a vital role in their properties of relevance to biology or medicinal chemistry[58–63], although major advances in transition-metal-catalyzed fluoroalkylation have been achieved in recent years[64–68]. It is worth noting that the elegant Ni-catalyzed tandem difluoroalkylation–(alkyl)arylation of enamides to the synthesis of difluoroalkylated amides was illustrated by Zhang and coworkers[69–71]. To the best of our knowledge, organozinc reagents for transition-metal-catalyzed difunctionalization of alkenes and 1,3-dienes to achieve fluorinated compounds were rather rare and limited to the use of nickel catalysis with activated alkenes[71]. In particular, the much less toxic and industrial-friendly cobalt catalysts have unfortunately thus far proven elusive for the aforementioned three-component cascade coupling reactions[72,73].

Recently, our laboratory reported the fluorine installation through cobalt-[74] and copper-catalyzed[75] alkyne and alkene difunctionalization strategies. In this work, we report a versatile cobalt-catalyzed regioselective difluoroalkylarylation of (un)activated alkenes and 1,3-dienes with polyfunctionalized arylzinc pivalates and difluoroalkyl bromides (Fig. 1b), which provides an expedient method to install fluorine into complex compounds. Of special interest in this cobalt catalysis is that the arylzinc pivalates seem very crucial for promoting the overall catalytic efficacy.

## Results

We initiated our studies by optimizing reaction conditions for the envisioned cobalt-catalyzed regioselective three-component coupling of alkenylarene (2a) with bromodifluoroacetate (1a) and phenylzinc pivalate (3a, PhZnOPiv). A cascade cross-coupling reaction was observed in the presence of 10.0 mol% CoBr$_2$ under ligand-free conditions, thus affording the desired aryl-difluoroalkylated product 4 in 83% yield with high regioselectivity (Fig. 2a, entry 1). Among a number of representative chelating ligands, bipyridines have given negative effects, and only a trace amount of product was observed (entries 2–3); tridentate 2,6-bis(N-pyrazolyl)pyridine, 1,10-phenanthrolines, diimine, TMEDA, ME$_4$DACH, as well as dppbz ligands gave poor-to-high yields, whereas the neocuproine (L5) afforded 4 in 92% (entries 4–10). Further, testing reactions with different solvents verified the crucial importance of MeCN as the reaction medium (entry 11; see SI). Switching from CoBr$_2$ to other representative cobalt salts, such as CoCl$_2$ and CoCl$_2$(PPh$_3$)$_2$, led to significant reduced yields (entry 12–13). In sharp contrast, replacement of CoBr$_2$ by using NiBr$_2$, FeCl$_2$, CrCl$_2$, or CuBr failed to furnish the desired product 4 (entry 14).

Lei[76–78] demonstrated that arylzinc reagents prepared by different methods possess very different kinetics in palladium- and nickel-catalyzed oxidative couplings, and further X-ray absorption spectroscopy studies show that changing the halide anion from Cl to Br or I will result in an increase of the Zn–C bond distance and thereby improve the trans-metalation rate[79]. In order to preliminarily reveal the different kinetics between this solid zinc reagent and conventional zinc reagents, a series of control experiments with six different phenylzinc reagents, which prepared by transmetallation of the corresponding phenylmagnesium halides and zinc halides (Note: PhZnOPiv was prepared from PhMgCl and Zn(OPiv)2 with 1:1.2 ratio; Ph2Zn·2Mg(OPiv)Cl was prepared from PhMgCl and Zn(OPiv)2 with 2:1 ratio; PhZnX was prepared from PhMgX and ZnX2 with 1:1.2 ratio; Ph2Zn·2MgCl2 was prepared from PhMgCl and ZnCl2 with 2:1 ratio), were also performed under the ligand-free cobalt catalysis (Fig. 2b). Interestingly, all of these reactions were almost finished within remarkably short reaction times of only 15 min. It is worth noting that significantly reduced conversions of 4 were observed when using PhZnX (X = Cl, Br, or I), Ph$_2$Zn·2MgCl$_2$, or Ph$_2$Zn·2Mg(OPiv)Cl instead of PhZnOPiv. Moreover, the results of comparison experiments between Ph$_2$Zn·2Mg(OPiv)Cl and Ph$_2$Zn·2MgCl$_2$ show the superiority of the former as well. Hence, these observations highlighted that the presence of M(OPiv)$_2$ (M = Mg or Zn) has made these organozinc pivalates stand out among salt-supported organometallics, thus displaying the distinct advantage of reacting well in our regioselective cobalt-catalyzed difluoroalkylarylation of olefins.

Subsequently, the versatility of this optimized cobalt(II) catalyst was examined in a range of difluoroalkylarylation reactions with various polyfunctionalized arylzinc pivalates 3 (Fig. 3a). All arylzinc pivalates were prepared from the corresponding aryl halides by Mg insertion in the presence of LiCl[80]. Although the neocuproine (L5) gave the optimal results in the model reaction, in our efforts to extend the substrate scope of this domino reaction, ligand-free CoBr$_2$ proved to be superior (see the results of products 7, 9, 11). A variety of para- and/or metal-substituted arylzinc pivalates were identified as viable nucleophiles for difluoroalkylarylation with bromodifluoroacetate (1a) and 4-methoxystyrene (2a) to afford the desired products 4–16 in moderate yields. More sterically hindered 4-chloro-2-

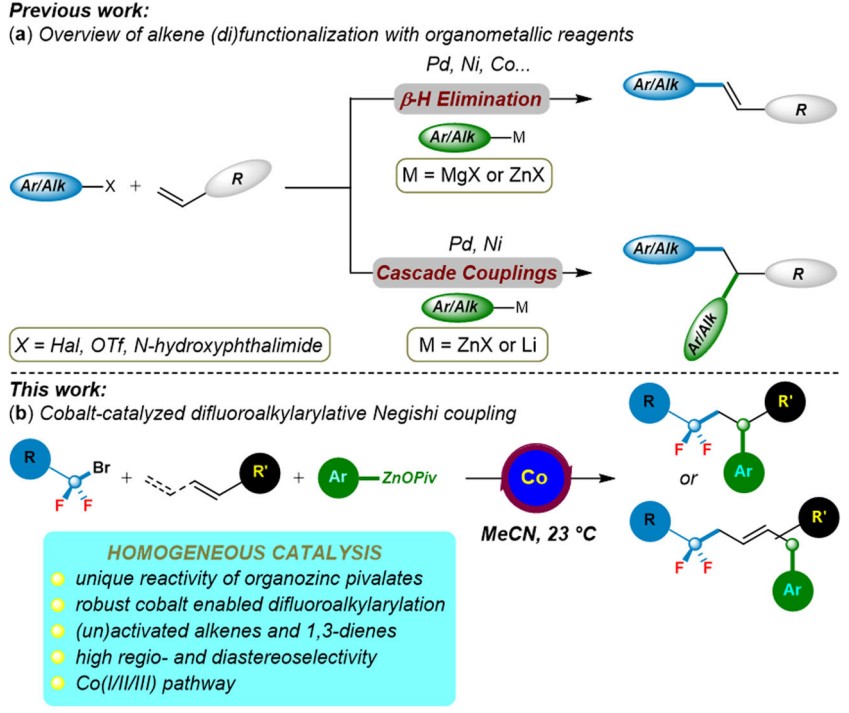

**Fig. 1 Background and our objective. a** Organometallic reagents for difunctionalization of olefins. **b** Organozinc pivalates for cobalt-catalyzed difluoroalkylarylation of olefins.

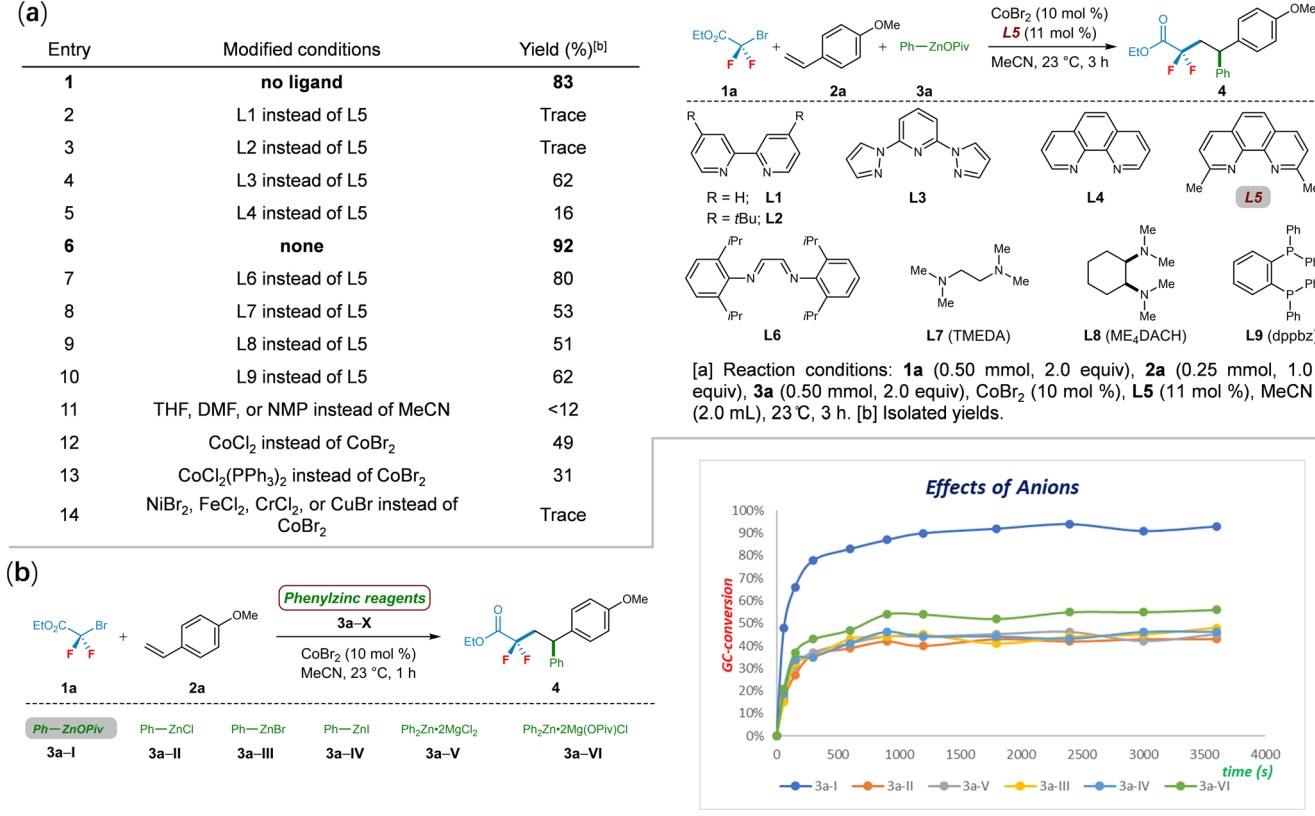

**(a)**

| Entry | Modified conditions | Yield (%)[b] |
|-------|---------------------|--------------|
| **1** | **no ligand** | **83** |
| 2 | L1 instead of L5 | Trace |
| 3 | L2 instead of L5 | Trace |
| 4 | L3 instead of L5 | 62 |
| 5 | L4 instead of L5 | 16 |
| **6** | **none** | **92** |
| 7 | L6 instead of L5 | 80 |
| 8 | L7 instead of L5 | 53 |
| 9 | L8 instead of L5 | 51 |
| 10 | L9 instead of L5 | 62 |
| 11 | THF, DMF, or NMP instead of MeCN | <12 |
| 12 | CoCl₂ instead of CoBr₂ | 49 |
| 13 | CoCl₂(PPh₃)₂ instead of CoBr₂ | 31 |
| 14 | NiBr₂, FeCl₂, CrCl₂, or CuBr instead of CoBr₂ | Trace |

[a] Reaction conditions: **1a** (0.50 mmol, 2.0 equiv), **2a** (0.25 mmol, 1.0 equiv), **3a** (0.50 mmol, 2.0 equiv), CoBr₂ (10 mol %), **L5** (11 mol %), MeCN (2.0 mL), 23 °C, 3 h. [b] Isolated yields.

**(b)**

**Fig. 2 Reaction optimization and reactivity of different arylzinc reagents. a** Optimization for cobalt-catalyzed difluoroalkylarylation. **b** Kinetic experiments with different phenylzinc reagents of 3a–X.

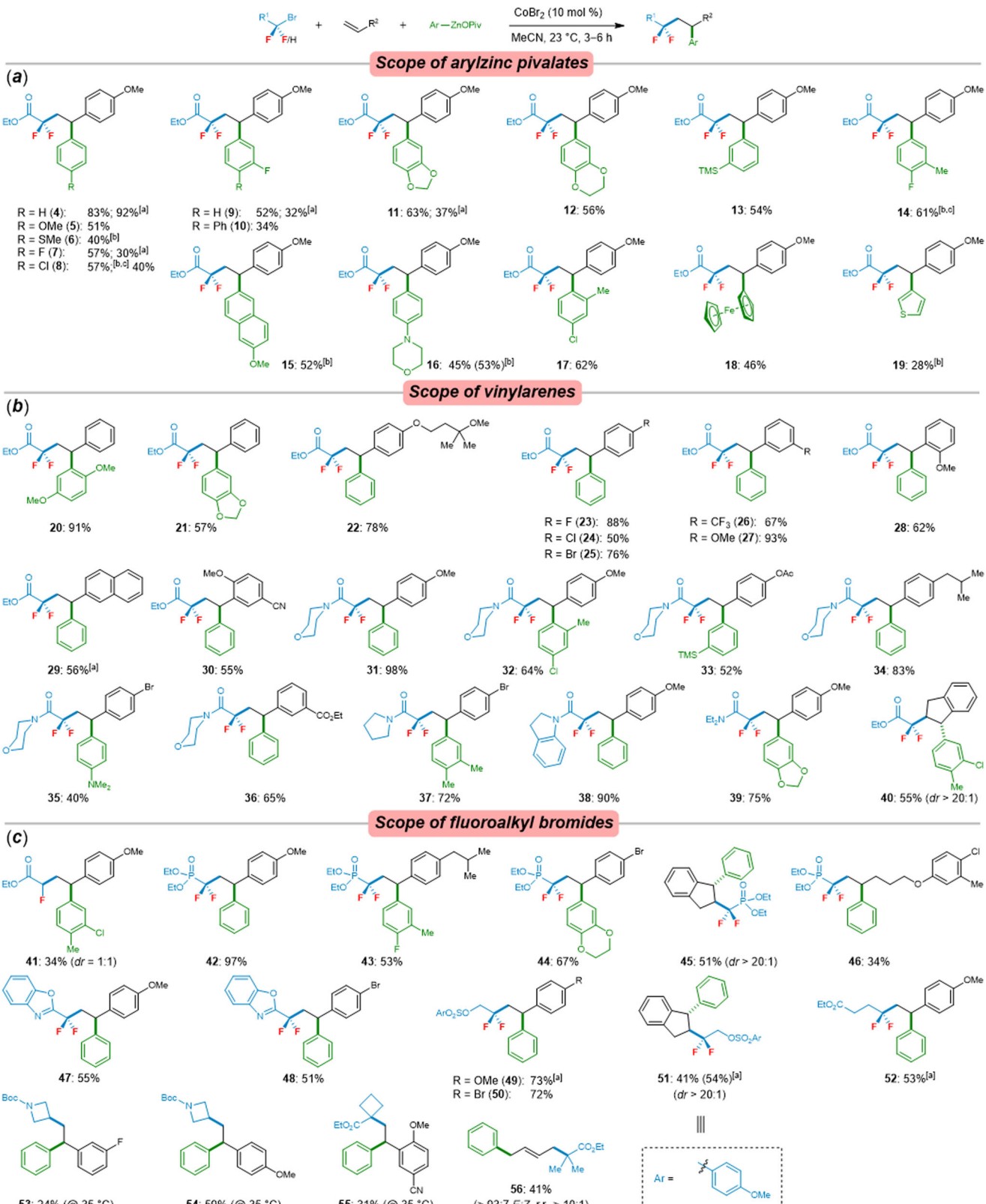

**Fig. 3 Scope of cobalt-catalyzed cascade cross-coupling reaction. a** Substrate scope of arylzinc pivalates. **b** Substrate scope of alkenylarenes and **c** fluoroalkyl bromides. Reaction conditions: alkyl bromides (0.50 mmol, 2.0 equiv), alkenes (0.25 mmol, 1.0 equiv), arylzinc pivalates (0.50 mmol, 2.0 equiv), CoBr₂ (10 mol %), and MeCN (2.0 mL), 23 °C, 3–6 h. [a] About 11 mol% of **L5** was used. [b] CoBr₂ (20 mmol%) was used. [c] Ar₂Zn·2Mg (OPiv)Cl (2.0 equiv) was used.

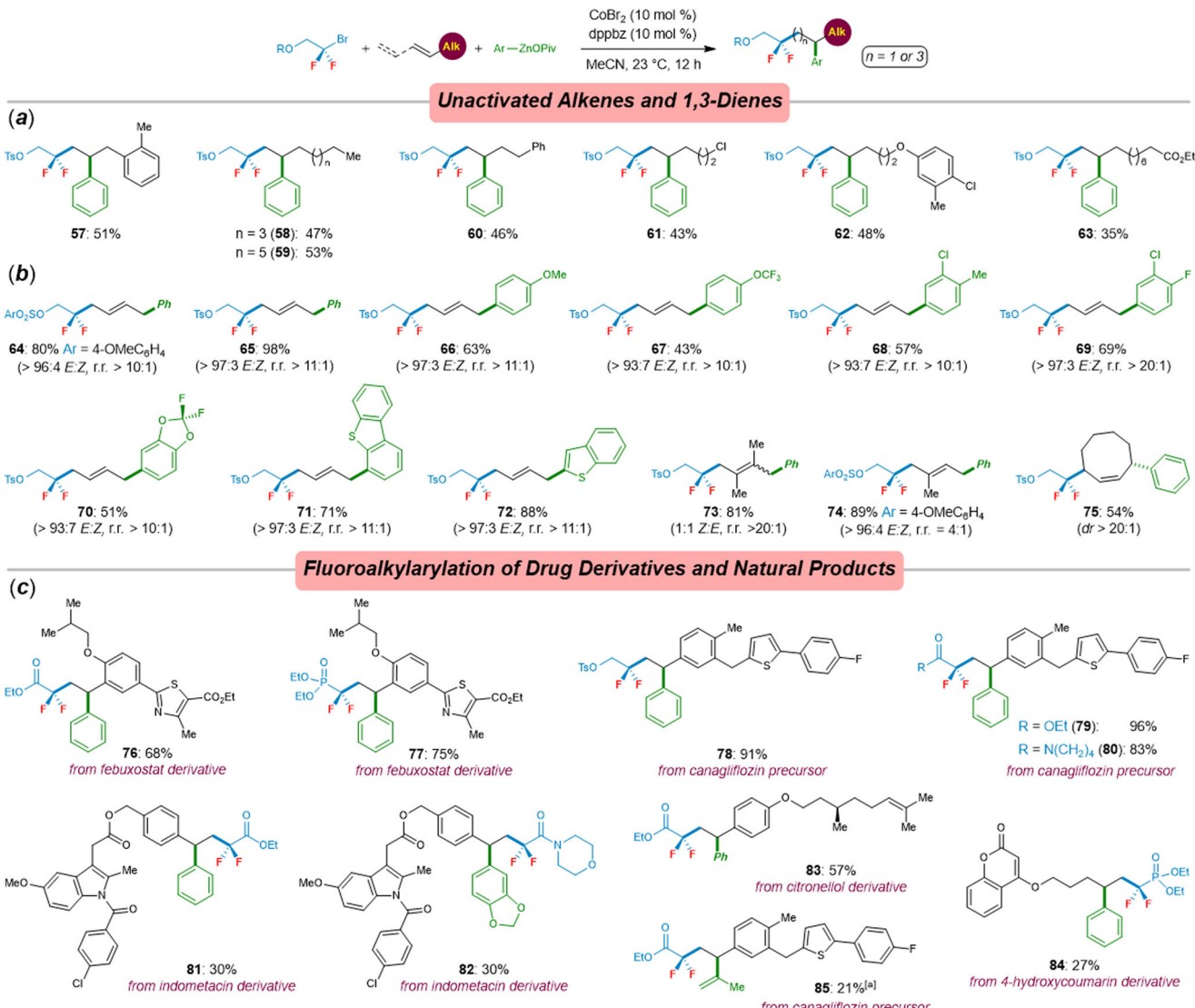

**Fig. 4 Difluoroalkylarylation of alkenes. a** Scope of unactivated alkenes. **b** Scope of 1,3-dienes. **c** Late-stage difluoroalkylarylation of drug derivatives and natural products. Reaction conditions: **1k** (0.25 mmol), unactivated alkenes or 1,3-dienes (0.5 mmol), arylzinc pivalates (0.5 mmol), CoBr₂ (10 mol%), dppbz (10 mol%), and MeCN, 23 °C, 12 h. [a] A second portion of **1a** and isopropenylzinc pivalate was added.

methylphenylzinc pivalate was successfully employed, leading to the desired difluoroalkylarylated product **17** in 62% yield. Notably, ferrocenylzinc pivalate, as well as 3-thienylzinc pivalate also smoothly underwent the cobalt-catalyzed cascade cross-coupling, albeit yielding the products **18-19** in relatively lower yields. Thus far, electron-withdrawing groups substituted arylzinc pivalates proven to be unsuitable nucleophiles.

Thereafter, we have explored the substrate scope of the difluoroalkylarylation reaction with a wide range of vinylarenes and bromodifluoroacetate/amides (Fig. 3b). Remarkably, alkenyl-larenes bearing various valuable electrophilic functional groups, such as ether (**22**), fluoro (**23**), chloro (**24**), bromo (**25, 35, 37**), trifluoromethyl (**26**), methyloxy (**27-28, 31-32, 38-39**), cyano (**30**), acetate (**33**), esters (**36**), and isobutyl (**34**) substituents, as well as vinylnaphthalene (**29**) and unsubstituted styrenes (**20-21**), were well tolerated under the reaction conditions and converted to the corresponding difluorinated 1,1-diarylalkanes in moderate-to-excellent yields (40-98%), as were also observed when using different bromodifluoroacetamides as the fluorinating reagents. Also, internal alkene with (E)-β-methylstyrene was examined under our cobalt catalysis, but only a trace amount of the desired

product was detected by GC analysis (see SI). In sharp contrast, coupling of arylzinc pivalate, bromodifluoroacetate with indene gave the desired difluoroalkylarylated product **40** in 55% yield, with high diastereoselctivity (dr > 20:1). In addition, we further investigated the cross-coupling of various fluoroalkyl bromides with olefins and arylzinc pivalates. In contrast to bromodi-fluoroacetate, the bromomonofluoroacetate only gave 34% yield under the standard reaction conditions, and with a poor dia-stereoselectivity (dr = 1:1) (Fig. 3c; **41**). We were also pleased to find that bromodifluoromethylphosphonate smoothly underwent the envisioned cobalt-catalyzed difluoromethylarylation to afford the desired 1,1-diarylalkylphosphonates **42-45** in 51-97% yields, and compound **45** was obtained with high diastereoselectivity (dr > 20:1). Besides, the unactivated alkene furnished the desired difluoromethylarylated phosphonate **46** as well, albeit in a modest yield. Additionally, using as substrate of α-bromodifluoromethyl-substituted benzoxazole proved to be viable with versatile cobalt catalyst and, thereby, provided **47-48** as the products in 51-55% yields. Remarkably, this cobalt-catalyzed regioselective difluor-oalkylarylation reaction was further extended to the decorated difluoroalkyl bromide. Functional groups, such as arylsulfonate,

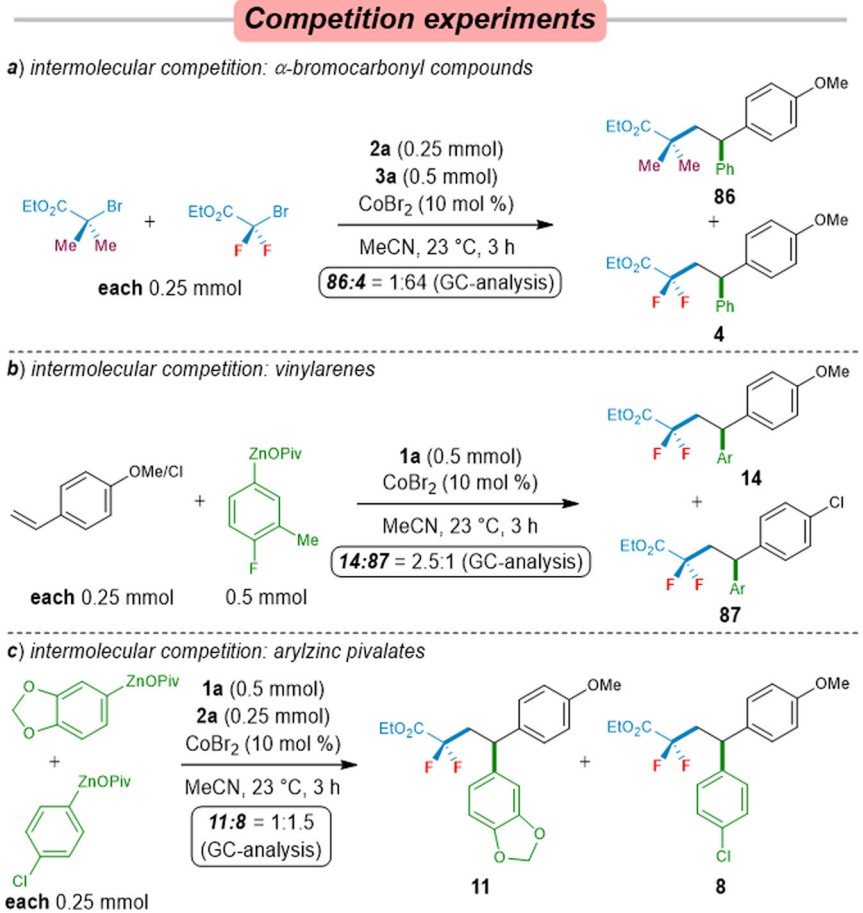

**Fig. 5 Control experiments. a** Competition experiments between different alkyl bromides. **b** Competition experiments between different vinylarenes. **c** Competition experiments between different arylzinc pivalates.

ester, were well tolerated under the standard reaction conditions, thus delivering the desired products **49–52** in good yields and with high diastereoselectivity of **51** (*dr* > 20:1). To our delight, the secondary and tertiary alkyl iodides were also successfully performed for the cobalt-catalyzed alkylarylation, albeit delivering the desired products **53–56** in 24–59% yields.

Transformations of unactivated alkenes are acknowledged widely as a challenge in transition-metal-catalyzed difunctionalization of alkenes[49–56,69–71]. The reaction conditions previously optimized for the alkenylarenes led to an unsatisfactorily low yield of **57**, because significant amounts of a Heck-type coupling product were formed as well. However, we were delighted to find that the transformation of the unactivated alkene difluoroalkylarylation process was significantly improved when using dppbz (**L9**, 10 mol%) as the ligand, leading to **57** in 54% yield. A number of unactivated alkenes were readily converted into the desired difluoroalkylarylated products **58–63** in moderate yields (See more details from the Supporting Information). Moreover, various synthetically valuable functional groups, including chloro, ether, and ester, remained intact by the cobalt catalyst (Fig. 4a). Beyond that, the possibility of cobalt-catalyzed difluoroalkylation to form an allyl radical, which subsequently underwent 1,3-shift and Csp³–Csp² cross-couplings with arylzinc pivalates, was also investigated (Fig. 4b). Indeed, difluoroalkyl bromide **1k** and a quite range of functionalized (hetero)aryl-zinc reagents were realized: 1,4-difunctionalization of 1,3-dienes with good regioselectivity and diastereoselectivity, thus furnishing **64–74** in 43–98% yields, albeit products **73** and **74** were obtained with 1:1 *E:Z* selectivity and 4:1 regioselectivity, respectively. To our delight,

1,3-octadiene was proven to be a suitable substrate as well, giving the product **75** with high diastereoselectivity.

To further illustrate the potential applications of this cobalt-catalyzed regioselective difluoroalkylarylation in late-stage functionalizations of pharmaceutically active molecules, alkenylarenes derivatized from (pre-)drug molecules, such as febuxostat, canagliflozin, as well as indomethacin, were well difluoroalkylarylated with arylzinc pivalates and α-bromodifluorocarbonyl compounds or bromodifluoromethylphosphonate, leading to the corresponding products **76–82** in 30–96% yields. These results show the potential utility of this protocol for the discovery of bioactive drugs. Importantly, citronellol derivative was readily incorporated into the product **83** with remarkably high regioselectivity and chemoselectivity. Moreover, an unactivated alkene bearing a 4-hydroxycoumarin proved to be a viable substrate as well, albeit delivering the phosphonate **84** in a rather modest yield. Finally, we showed that isopropenylzinc pivalate is well suited for the cobalt-catalyzed difluoroalkylalkenylation, although the reaction proceeded with lower yield (Fig. 4c).

Intrigued by the high regioselectivity and efficacy of our cobalt-catalyzed difluoroalkylarylation, a series of intermolecular competition experiments were performed (Fig. 5). A competition experiment between bromodifluoroacetate (**1a**) and 2-bromo-2-methylpropanoate showed that BrCF₂CO₂Et reacted much faster than these α-bromocarbonyl compounds. These findings can be rationalized in terms of a prioritized direct halogen atom abstraction from difluoroalkyl bromides via single-electron transfer from a cobalt catalyst (Fig. 5a)[81]. Intermolecular competition experiments with different alkenylarenes and arylzinc

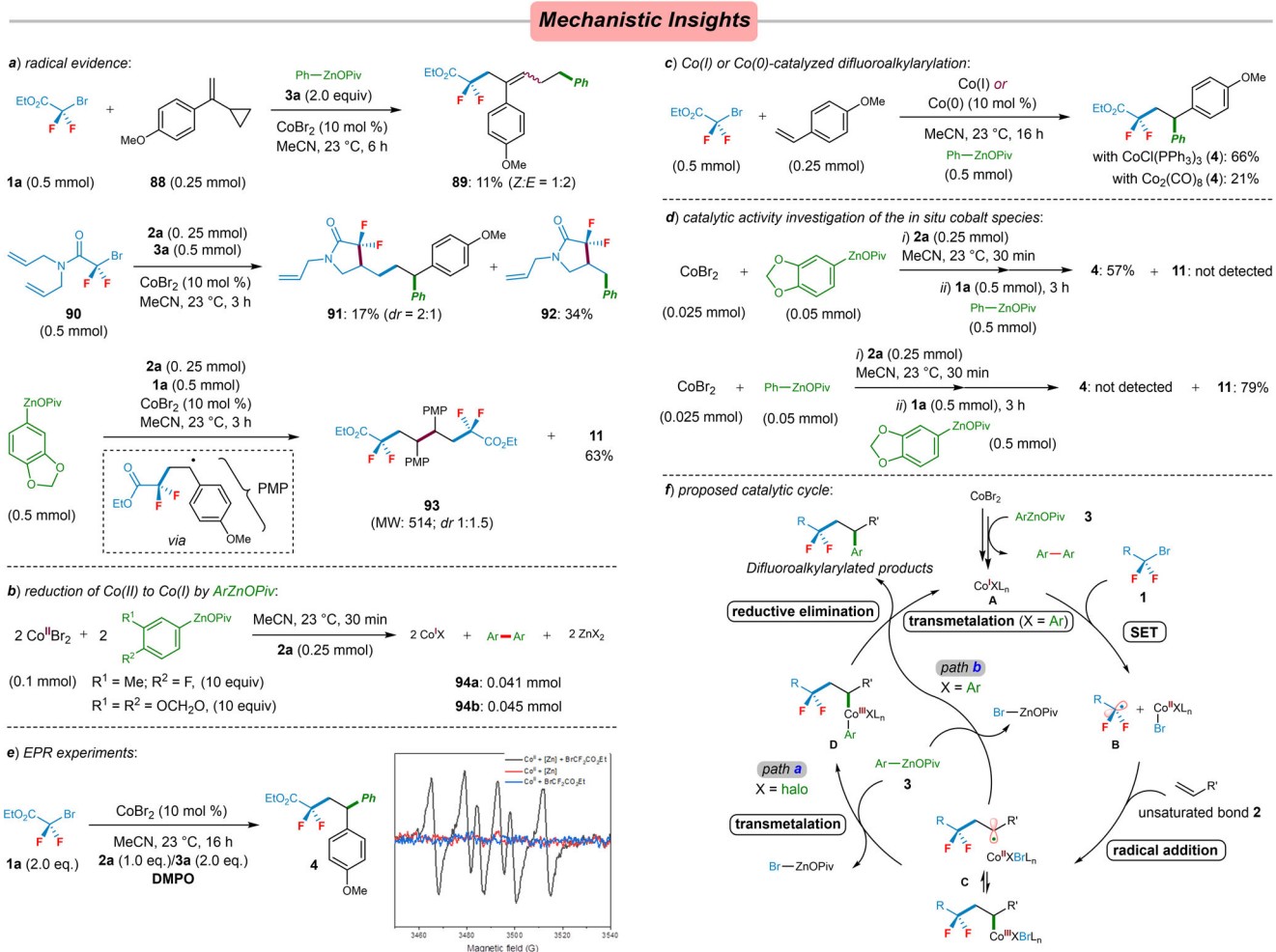

**Fig. 6 Mechanistic studies for cobalt-catalyzed cascade difluoroalkylarylation. a** Radical evidences. **b** Reduction of CoBr₂ by arylzinc pivalates. **c** Reactivity of well-defined Co(I), Co(0) catalysts. **d** Catalytic activity investigation of the in situ low-valent cobalt species. **e** EPR experiments with DMPO (DMPO = 5,5-dimethyl-1-pyrroline-1-oxide).

pivalates revealed electron-rich styrenes and electron-deficient arylzinc pivalates to be slightly reactive substrates (Fig. 5b, c). These results suggested that vinylarenes and arylzinc reagents might not be involved in the rate-determined step[56].

Beyond that, the radical-clock experiment with substrate **88** bearing a radical clock cyclopropane moiety, the ring-opened difluoroalkylarylated product **89**, was generated in 11% yield. Similarly, both three- and two-component coupling products were observed when using *N,N*-diallyl-2-bromo-2,2-difluoroacetamide (**90**) as a radical probe under the standard reaction conditions, the cyclized products **91** (*dr* = 2:1) and **92** were generated in 17% and 34% yields, respectively. Moreover, a difluoroalkylated benzylic radical homocoupling dimer **93** was detected by GC as well. With these findings, we propose that this cobalt-catalyzed difluoroalkylarylation involves a single-electron-transfer (SET) process (Fig. 6a).

According to the earlier mechanistic studies for cobalt-catalyzed cross-coupling reactions with using organomagnesium reagents, an in situ low-valent Co(0) was proposed as the catalytically active species[52,72,73,82,83]. On the other hand, a mechanism involving Co(I)/(III) couple was also proposed for many cobalt-catalyzed cross-couplings[37–39,81]. Therefore, we performed experiments of CoBr₂ (1.0 equiv) with excess of ArZnOPiv under typical reaction conditions for 30 min. These reactions furnished the corresponding

homoproducts of **94a** and **94b** in near-0.5 equiv ratio to that of CoBr₂, respectively. These findings support the formation of a Co(I) species based on the stoichiometry shown in Fig. 6b. In this context, the well-defined Co(I) complex, such as CoCl(PPh₃)₃ was proved to be active for the desired difluoroalkylarylated process, yielding product **4** in 66%, while Co₂(CO)₈ gave a poor yield (Fig. 6c). Further experiments to examine the catalytic activity of the in situ-generated low-valent cobalt(I) species were performed. A mixture of vinylarene **2a** (0.25 mmol) and CoBr₂ (0.025 mmol) was treated with 2.0 equiv of 3,4-(methylenedioxy)phenylzinc pivalate (0.05 mmol) at 23 °C for 30 min to generate the proposed Co(I) species, followed by addition of bromodifluoroacetate **1a** (0.3 mmol) and another 0.5 mmol of phenylzinc pivalate. The difluoroalkylarylated product **4** was isolated in 57% yield as the sole product, while the product **11** was obtained in 79% yield when exchanging the order of the two arylzinc reagents (Fig. 6d). These findings are consistent with the in situ-generated low-valent cobalt(I) species that might be the active catalyst for the current three-component cross-coupling reaction. A series of EPR spin-trapping experiments show the existence of *C*-centered radicals trapped by DMPO (*g* = 2.0066, A_N = 13.9 G, A_H = 19.3 G), which was considered to be •CF₂R[84]. These results strongly supported that the single-electron-transfer process for the activation of BrCF₂R was only promoted by the in situ-formed Co(I) species (Fig. 6e).

**Fig. 7 Applications. a** Gram-scale reaction. **b** Modification of difluoroalkylarylated products.

Based on the above experimental findings, along with previous mechanistic insights[37–39,74,81], a mechanism for this regioselective cobalt-catalyzed difluoroalkylarylation of alkenes has been proposed as shown in Fig. 6f. The reduction of the precatalyst CoBr$_2$ with arylzinc pivalates forms the catalytically active Co(I) species (**A**), which reduces difluoroalkyl bromides (**1**) by SET and generates difluoroalkyl radical **B**, then followed by a facile radical addition of **B** into olefins (**2**) to afford a secondary alkyl radical species, along with subsequent rapid trapping with L$_n$Co(II)XBr (X = Br) into intermediate **C**, which undergoes trans-metalation with ArZnOPiv (**3**) to lead to the organocobalt(III) species **D**. Subsequent reductive elimination finally delivers the difluoroalkylarylated product and regenerates the active cobalt(I) catalyst (path **a**). In addition, another possible pathway is that transmetallation of arylzinc pivalates could also occur after the initial reduction step, thus in situ forming the L$_n$Co(I)X (X = Ar) species as the catalyst to promote the SET process. Radical addition and reductive elimination give rise to the desired products and regenerate the active Co(I) species (path **b**).

We were also pleased to find that this cobalt-catalyzed difluoroalkylarylation can be easily scaled up to gram level. Under the optimized reaction conditions, the difluoroalkylarylated product **95** was afforded with high efficacy (65% yield, Fig. 7a). Finally, we further demonstrated the synthetic potential of this cobalt-catalyzed difluoroalkylarylation strategy through the late-stage modification of the obtained difluoroalkylarylated products. For example, the resulting N-morpholino amide **95** can be readily converted into various ketones by treating with Grignard reagents, thus furnishing the products **97a**–**b** in moderate yields. Moreover, the reduction of the ester group of substrate **4** by using NaBH$_4$ provides the corresponding alcohol **98**, which readily undergoes various derivatizations (Fig. 7b).

## Discussion

In conclusion, we have reported the practical cobalt catalysis for regioselective difluoroalkylarylation of alkenes or 1,3-dienes with functionalized arylzinc pivalates and difluoroalkyl bromides. This simple cobalt catalyst enables three-component cross-couplings through cascade Csp$^3$–Csp$^3$/Csp$^3$–Csp$^2$ bond formation in one-pot fashion, thus generating difluoroalkylarylated products with predictable regioselectivity and high diastereoselectivity. The reaction proceeds under remarkable mild conditions with high efficacy, excellent functional group tolerance, as well as a broad substrate scope. Notable features of this approach are the use of less toxic and low-cost cobalt catalyst, as well as user-friendly solid zinc reagents. Straightforward late-stage functionalizations of pharmaceutically active molecules show the potential applications of this protocol for the discovery of bioactive drugs. Beyond that, among a series of kinetic experiments with six types of phenylzinc reagents, these solid arylzinc pivalates displayed the distinct advantage of reactivity for the current reaction. Detailed mechanistic studies demonstrated that the reaction undergoes a direct halogen atom abstraction via single-electron transfer from the in situ-formed cobalt(I) species to difluoroalkyl bromides.

## Methods

**Cobalt-catalyzed difluoroalkylarylation of alkenes**. A suspension of CoBr$_2$ (10 mol%), olefin (0.25 mmol, 1.0 equiv), difluoroalkyl bromide (0.5 mmol, 2.0 equiv), and aryl zinc pivalates (0.5 mmol, 2.0 equiv) in degas MeCN (1.0 mL) was stirred at 23 °C for 3 h under an atmosphere of Ar. At ambient temperature, the solvent was evaporated under reduced pressure and the remaining residue was purified by column chromatography on silica gel to yield the desired products.

## Data availability

The authors declare that all other data supporting the findings of this study, including experimental procedures and compound characterization, are available within the article and its Supplementary Information files.

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

## Acknowledgements

We thank the National Natural Science Foundation of China (21602083) and Start-up Grant of Soochow University (GJ10900220) for financial support. We are grateful to Prof. Yi-hung Chen for their kind discussions, and Jianlin Yao, and Chen Zhu for their kind support and accessing to the laboratory.

## Author contributions

X. C., X. L., and S. W. contributed equally to this work and they planned, conducted, and analyzed the experiments. X. C., X. L., S. W., Y. H., and B. H. performed the experiments. J. L., and A. L. designed and directed the project and wrote the paper with contributions from all coworkers.

## Competing interests

The authors declare no competing interests.
