## [Peer Review File · Nature Communications]

Reviewers' Comments:

Reviewer #1:

Remarks to the Author:

This manuscript describes a cobalt-catalyzed difluoroalkylarylation of alkenes. Various kinds of difluoroalkyl bromide can be used in this transformation and bench-stable solid arylzinc reagent is discovered and used. This three-component coupling reaction use less toxic and low-cost cobalt catalyst, and shows high regio- and distereoselectivity. The substrate scope reveals a good toleration of activated, unactivated alkenes and 1,3-dienes. Several pharmaceutically active molecule derivatives have be applied successfully for late-stage functionalizations using this interesting method. A number of mechanistic studies support that a single electron transfer occurs between in situ formed Co(I) species and difluoroalkyl bromides. Based on these, the reviewer recommend this work to be published on Nature Communications after minor revision, and some comments/suggestions are shown below:

1. The authors underlined the bench-stable solid arylzinc pivalates. It was suggested that the storage method should be given in the manuscript or SI.
2. None of the electron-poor arylzinc reagent was used in this paper. Whether this kind of arylzinc pivalate is difficult to synthesize or is not tolerated under the standard catalytic conditions. Please comment this limitation in the manuscript.
3. Line 162, Page 9, the authors declaimed the reaction of 1,3-diene "underwent 1,3-H-shift" process. Could the authors explain which hydrogen atom shift occurs? It looks like this process is only a resonance of allyl radical.
4. In the view of product 51 and 71, does this difluoroalkylarylation have chemoselectivity of internal and inner alkenes?
5. Some amounts of the reactants are not clearly in the mechanistic research. For example, Scheme 8a, b, 9a.
6. In Scheme 11b, the reduction of ester required 15 equiv. of NaBH₄. Could the authors use a fewer amount of the reductant to give a bit of lower yield, or elevated the reduction ability via adding AlCl₃, etc.?

Reviewer #2:

Remarks to the Author:

The manuscript by Li et. al describes three-component Co-catalyzed difluoroalkylarylation of alkenes along with some mechanistic insights. The condition employed for this transformation involves the use of 10 mol % of CoBr₂ as a catalyst to prepare a wide range of difluoroalkylarylated in moderate to good yield. As a part of mechanistic investigation, the authors performed kinetic studies and competition experiments to shed light on the mechanism. The foundation of the current work lies on the literature report including *Org. Lett.* 2021, 5387-5391 by the same authors along with several other seminal reports (*Angew. Chem. Int. Ed.* 2016, 55, 12270; *JACS* 2018, 140, 9801; *Angew. Chem. Int. Ed.* 2019, 8047 etc.) that illustrates both two component and three-component reactions identical to the current reaction developed by the authors. In light of the lack of novelty and that the scope is also seriously limited to α,α -difluoroesters, it's my opinion that this work is not suitable for Nature Communications. It's appropriate for more specialized journals like *Org. Lett.*, *Eur. J. Org. Chem.* etc.

1. Several compounds are not pure and the H and C spectra are not in publishable quality: compounds 6, 7, 8, 10, 14, 29, 40, 44, 48, 50, 53 (after this I stopped looking).
2. One of the serious limitations of the work is its scope - the authors should include examples in which alkyl bromides and iodides could also be used as a coupling partner, not just simple difluoro ester derivatives.
3. The authors did not provide any justification on the use of high catalyst loading.
4. The reaction already works so well without any ligand (Table 1, entry 1). They why both to add any ligand? This increase in yield is marginal and could be just within the experimental error.
5. I have a serious concern over the use of M(OiPiv)₂ appendaged with phenylzinc reagent. What's so special about this moiety as it resembles vey closely with the analogous PhZnOPiv?
6. Despite the use of high catalyst loading along with L5, the yield of some of the synthesized product seem quite low. For example, 37% in case of compound 14. What is the mass balance? Are there any side products?

7. The authors should display the synthetic utility of some other fluorinating reagents other than the standard bromodifluoroester derivative used.

8. Comment on diastereoselectivity: It seems like a good (10:1) is reported for several analogs however some discrepancy also prevails. The authors should provide a good structure-based model that could clearly explain the outcome of high diastereoselectivity.

Reviewer #3:

Remarks to the Author:

The manuscript by Li and coworkers developed an efficient method to prepare difluoroalkylated compounds through cobalt-catalyzed aryldifluoroalkylation of alkenes. The reaction uses arylzinc pivalates as one of the coupling partners, enabling the tandem reaction to allow a variety of alkenes, including styrenes, 1,4-dienes, and unactivated aliphatic alkenes. This is a good extension to the previous work reported by Zhang (refs. 69-71), and the work overcomes the limitations that a chelating group is needed in the previous work, thus expanding the substrate scope to prepare diversified difluoroalkylated compounds with different alkenes. To demonstrate the utility of this protocol, the modification of complex bioactive molecule derivatives were conducted. The authors also conducted detailed mechanistic studies, and a Co(I/II/III) catalytic cycle was proposed, which is reasonable. Overall, this is a nice work, and the paper can be published in Nature Communications after minor revision.

- 1) For the reaction with unactivated aliphatic alkenes, only $\text{ArO}_2\text{SOCH}_2\text{CF}_2\text{Br}$ were applicable to the reaction. How about other difluoroalkyl bromides, such as $\text{BrCF}_2\text{CO}_2\text{Et}$, $\text{BrCF}_2\text{alkyl}$? If these substrates were not applicable to the reaction, one of the reasons is the $\text{S}=\text{O}$ on $\text{ArO}_2\text{SO}-$ moiety functionalizes as the chelating group to chelate with cobalt, thus facilitating the catalytic cycle.
- 2) The path b in the catalytic cycle should be revised, as the transmetallation between cobalt(I) and arylzinc reagent is missed in the catalytic cycle.

Specific answers to the *Reviewers*

Reviewer #1 (Remarks to the Author):

Comments: This manuscript describes a cobalt-catalyzed difluoroalkylation of alkenes. Various kinds of difluoroalkyl bromide can be used in this transformation and bench-stable solid arylzinc reagent is discovered and used. This three-component coupling reaction use less toxic and low-cost cobalt catalyst, and shows high regio- and distereoselectivity. The substrate scope reveals a good toleration of activated, unactivated alkenes and 1,3-dienes. Several pharmaceutically active molecule derivatives have be applied successfully for late-stage functionalizations using this interesting method. A number of mechanistic studies support that a single electron transfer occurs between in situ formed Co(I) species and difluoroalkyl bromides. Based on these, the reviewer recommend this work to be published on Nature Communications after minor revision, and some comments/suggestions are shown below:

Response: Firstly, we thank this referee's kind comments.

Q1. The authors underlined the bench-stable solid arylzinc pivalates. It was suggested that the storage method should be given in the manuscript or SI.

Response: These fresh prepared arylzinc pivalates can be stored in ampoules filled with inert gas; we have added this message in the revised supporting information.

Q2. None of the electron-poor arylzinc reagent was used in this paper. Whether this kind of arylzinc pivalate is difficult to synthesize or is not tolerated under the standard catalytic conditions. Please comment this limitation in the manuscript.

Response: We also performed the reactions with the ethyl benzoate or benzonitrile-derived zinc pivalates; unfortunately, only trace amount of desired products were detected. Therefore, we have added the comments "Unfortunately, *para*-ester or nitrile substituted arylzinc pivalates proven to be unsuitable nucleophiles" in the revised manuscript.

Q3. Line 162, Page 9, the authors declaimed the reaction of 1,3-diene "underwent 1,3-H-shift" process. Could the authors explain which hydrogen atom shift occurs? It looks like this process is only a resonance of allyl radical.

Response: We thank this referee's valuable comments, and the description has been corrected in the revised manuscript with the following sentence: "the possibility of cobalt-catalyzed difluoroalkylation to form an allyl radical, which subsequently underwent 1,3-shift and Csp³-Csp² cross-couplings with arylzinc pivalates was also investigated".

Q4. In the view of product 51 and 71, does this difluoroalkylation have chemoselectivity of internal and inner alkenes?

Response: Substrate of 1*H*-indene has been utilized in different transition metal-catalyzed radical cross-coupling reactions, as reported by Zhang, Giri et al. (Angew. Chem., Int. Ed. **2020**, 59, 18741; **2020**, 59, 8047; **2013**, 52, 2529). The corresponding transformations always feature good chemoselectivity and high diastereoselectivity. However, β -methylstyrene did not work under this cobalt-catalysis.

Q5. Some amounts of the reactants are not clearly in the mechanistic research. For example, Scheme 8a, b, 9a.

Response: The detailed reaction conditions have been added in the revised manuscript.

Q6. In Scheme 11b, the reduction of ester required 15 equiv. of NaBH₄. Could the authors use a fewer amount of the reductant to give a bit of lower yield, or elevated the reduction ability via adding AlCl₃, etc.?

Response: We have repeated this reaction using 2.0 equiv. of NaBH₄ as the reductant, and the product **94** was obtained in 92%. We have added these information in the revised manuscript and supporting information.

Reviewer #2 (Remarks to the Author):

Comments: The manuscript by Li et. al describes three-component Co-catalyzed difluoroalkylarylation of alkenes along with some mechanistic insights. The condition employed for this transformation involves the use of 10 mol % of CoBr₂ as a catalyst to prepare a wide range of difluoroalkylarylated in moderate to good yield. As a part of mechanistic investigation, the authors performed kinetic studies and competition experiments to shed light on the mechanism. The foundation of the current work lies on the literature report including Org. Lett. 2021, 5387-5391 by the same authors along with several other seminal reports (Angew. Chem. Int. Ed. 2016, 55, 12270; JACS 2018, 140, 9801; Angew. Chem. Int. Ed. 2019, 8047 etc.) that illustrates both two component and three-component reactions identical to the current reaction developed by the authors. In light of the lack of novelty and that the scope is also seriously limited to α,α -difluoroesters, it's my opinion that this work is not suitable for Nature Communications. It's appropriate for more specialized journals like Org. Lett., Eur. J. Org. Chem. etc.

Response: Firstly, we thank this referee's comments.

Q1. Several compounds are not pure and the H and C spectra are not in publishable quality: compounds 6, 7, 8, 10, 14, 29, 40, 44, 48, 50, 53 (after this I stopped looking).

Response: We have tried our best to purify most of these compounds and improve the quality of spectra in the revised supporting information.

Q2. One of the serious limitations of the work is its scope - the authors should include examples in which alkyl bromides and iodides could also be used as a coupling partner, not just simple difluoro ester derivatives.

Response: We thank the referee's comments, we have also tried many experiments for the cobalt-catalyzed alkylarylation reaction. However, only tertiary alkyl iodides underwent the envisioned cross-coupling process, albeit giving the product in poor conversions. Please see more details below:

Entry	Modified conditions	P : Yield (%) ^[b]
1	L1-L6 or L9 instead of L7, DMF instead of MeCN	trace
2	L7 or L8, DMF instead of MeCN	10 or 8%
3	none	24%
4	PhZnCl instead of PhZnOPiv	11%
5	Co(acac) ₂ , CoBr ₂ , CoI ₂ , CoCl ₂ (PPh ₃) ₂ instead of CoCl ₂	15-30%
6	L9	trace
7	PhMe, THF, DMA, or NMP instead of MeCN	trace

[a] Reaction conditions: *t*BuI (0.50 mmol, 2.0 equiv), **2a** (0.25 mmol, 1.0 equiv), **3a** (0.50 mmol, 2.0 equiv), CoCl₂ (10 mol %), L7 (11 mol %), MeCN (1.5 mL), 23 °C, 6 h. [b] NMR yields.

Moreover, secondary and primary alkyl halides were also examined under the reaction conditions, while only gave negative results.

Unsuccessful substrates:

These results demonstrated that bulky alkyl iodides, such as *t*BuI, could undergo cobalt-catalyzed alkylarylation process, albeit delivering the desired product in poor conversion.

Q3. The authors did not provide any justification on the use of high catalyst loading.

Response: In four examples (14-16, 18), better yields were recognized when increasing the CoBr₂ loading.

Q4. The reaction already works so well without any ligand (Table 1, entry 1). They why both to add any ligand? This increase in yield is marginal and could be just within the experimental error.

Response: The desired product was isolated in 83% yield without using ligand, it is a good yield, but still has some starting material of 4-methoxystyrene **2a** left. Using 11 mol % of ligands to increase the conversion and afford a yield up to 92% should be understood in this context.

Q5. I have a serious concern over the use of M(OiPiv)₂ appendaged with phenylzinc reagent. What's so special about this moiety as it resembles very closely with the analogous PhZnOPiv?

Response: Professor Mulvey and Knochel firstly demonstrated the detailed structural insights of RZnOPiv·Mg(OPiv)X·nLiCl in 2014 (Angew. Chem., Int. Ed. **2014**, 53, 2706). In this reference, the results support that Mg(OPiv)₂ can enhance this stability by mopping up OH or O₂ antagonists and capturing and holding on to H₂O molecules, making them less accessible for hydrolyzing C–Zn bonds. On the other hand, X-ray absorption spectroscopy studies show that changing the halide anion from Cl to Br or I will result in an increase of the Zn–C bond distance and thereby improve the transmetalation rate, as was reported by Lei (Chem. Commun. 2013, 49, 9615; J. Am. Chem. Soc. 2010, 132, 9607). In this context, we believe that the presence of M(OPiv)₂ (M = Mg or Zn) has changed the Zn–C bond distance of these new organozinc pivalates, thus displaying the distinct advantage of reacting well in this cobalt-catalyzed difluoroalkylation.

Q6. Despite the use of high catalyst loading along with L5, the yield of some of the synthesized product seem quite low. For example, 37% in case of compound 14. What is the mass balance? Are there any side products?

Response: In the case of compound **14**, the isolated yield is 61%. Besides that, 32% of 4-methoxystyrene **2a** was recovered.

Q7. The authors should display the synthetic utility of some other fluorinating reagents other than the standard bromodifluoroester derivative used.

Response: We thank this referee's suggestions, the following difluoroalkylation reaction was performed using difluoroethyl bromide as the electrophile; the desired product was obtained in 91% yield. We have also added these results in the revised manuscript and supporting information.

Q8. Comment on diastereoselectivity: It seems like a good (10:1) is reported for several analogs however some discrepancy also prevails. The authors should provide a good structure-based model that could clearly explain the outcome of high diastereoselectivity.

Response: Substrate of 1*H*-indene has been utilized in different transition metal-catalyzed radical cross-coupling reactions, as reported by Zhang, Giri et al. (*Angew. Chem., Int. Ed.* **2020**, 59, 18741; **2020**, 59, 8047; **2013**, 52, 2529). Generally, the products always feature high diastereoselectivity in these reactions.

Reviewer #3 (Remarks to the Author):

Comments: The manuscript by Li and coworkers developed an efficient method to prepare difluoroalkylated compounds through cobalt-catalyzed aryldifluoroalkylation of alkenes. The reaction uses arylzinc pivalates as one of the coupling partners, enabling the tandem reaction to allow a variety of alkenes, including styrenes, 1,4-dienes, and unactivated aliphatic alkenes. This is a good extension to the previous work reported by Zhang (refs. 69-71), and the work overcomes the limitations that a chelating group is needed in the previous work, thus expanding the substrate scope to prepare diversified difluoroalkylated compounds with different alkenes. To demonstrate the utility of this protocol, the modification of complex bioactive molecule derivatives were conducted. The authors also conducted detailed mechanistic studies, and a Co(I/II/III) catalytic cycle was proposed, which is reasonable. Overall, this is a nice work, and the paper can be published in *Nature Communications* after minor revision.

Response: Firstly, we thank this referee's comments.

Q1. For the reaction with unactivated aliphatic alkenes, only ArO₂SOCH₂CF₂Br were applicable to the reaction. How about other difluoroalkyl bromides, such as BrCF₂CO₂Et, BrCF₂alkyl? If these substrates were not applicable to the reaction, one of the reasons is the S=O on ArO₂SO- moiety functionalizes as the chelating group to chelate with cobalt, thus facilitating the catalytic cycle.

Response: We thank this referee's suggestions. The following experiments were performed under the standard reaction conditions. Firstly, bromodifluoroacetate (**1a**) underwent the difluoroalkylation process and delivered the desired product **95** in 61% yield. However, the difluoroalkyl bromide (**1m**), which bearing a strong bidentate of 8-aminoquinoline, failed to deliver the three-component coupling product. These results might suggest the S=O on ArO₂SO- moiety functionalizes are not the chelating group. We have also added these experiments in the revised supporting information.

Q2. The path b in the catalytic cycle should be revised, as the transmetalation between cobalt(I) and arylzinc reagent is missed in the catalytic cycle.

Response: We thank this referee's suggestion. We have highlighted the transmetalation between cobalt(I) and arylzinc reagent in Scheme 10. In path b, the initial active catalyst is Ar–Co(I)L_n, which reduces difluoroalkyl bromides (**1**) by SET, followed by radical addition, transmetalation, and reductive elimination afford the desired product, and regenerate the active Ar–Co(I)L_n species.

Reviewers' Comments:

Reviewer #1:

Remarks to the Author:

The revised manuscript has answered all questions this reviewer raised, and thus could be accepted for publication on Nature Communications. Some suggestions could be considered during the final revision.

1) Considering the storage conditions for arylzinc pivalates, "bench-stable" was suggested to be removed from the manuscript.

2) While the electron-poor arylzinc reagents were not tolerated in this reaction, the added comments "Unfortunately, para-ester or nitrile substituted arylzinc pivalates proven to be unsuitable nucleophiles" should be revised to all electron-withdrawing groups at the phenyl ring.

Reviewer #2:

Remarks to the Author:

In my previous comments, I had two major concerns - 1) the limited scope of the reaction that only works with 1,1-difluorobromo compounds. The scope still remains the same (the new compound 74 still comes from 1,1-difluorobromo), and 2) quite a few products have a large amounts of impurities to the point of being not publishable. Although the authors state that they tried their best, both ¹H and ¹³C NMR spectra of all the compounds in question still look the same with a lot of impurities. Therefore, I could not, in my conscience, agree for its publication in Nat. Commun. without these major concerns being addressed.

Reviewer #3:

Remarks to the Author:

The authors have fulfilled the referees' requests. The paper can be published in Nat. Commun..

Reviewer #1 (Remarks to the Author):

Comments: The revised manuscript has answered all questions this reviewer raised, and thus could be accepted for publication on Nature Communications. Some suggestions could be considered during the final revision.

Response: Firstly, we thank this referee's kind comments.

Q1. Considering the storage conditions for arylzinc pivalates, "bench-stable" was suggested to be removed from the manuscript.

Response: We have deleted the word of "bench-stable" from the manuscript.

Q2. While the electron-poor arylzinc reagents were not tolerated in this reaction, the added comments "Unfortunately, para-ester or nitrile substituted arylzinc pivalates proven to be unsuitable nucleophiles" should be revised to all electron-withdrawing groups at the phenyl ring.:

Response: We have revised the text with the following sentence: "*Thus far, electron-withdrawing groups substituted arylzinc pivalates proven to be unsuitable nucleophiles.*" in the revised manuscript.

Reviewer #2 (Remarks to the Author):

Comments: In my previous comments, I had two major concerns - 1) the limited scope of the reaction that only works with 1,1-difluorobromo compounds. The scope still remains the same (the new compound 74 still comes from 1,1-difluorobromo), and 2) quite a few products have a large amounts of impurities to the point of being not publishable. Although the authors state that they tried their best, both ^1H and ^{13}C NMR spectra of all the compounds in question still look the same with a lot of impurities. Therefore, I could not, in my conscience, agree for its publication in Nat. Commun. without these major concerns being addressed.

Response: Firstly, we thank this referee's comments.

On the basis of this reviewer's suggestions, several secondary and tertiary alkyl iodides have been utilized for the desired cobalt-catalyzed alkylarylation reaction. Please see more details below scheme 1. These results demonstrated that bulky alkyl iodides, such as 1-boc-3-iodoazetidione, 2-Iodo-2-methylpropionate and 1-iodocyclobutane-1-carboxylate could undergo cobalt-catalyzed alkylarylation process, albeit delivering the desired product in 22–59% yields. We have added these results in the revised manuscript and supporting information. However, *t*BuI, *i*PrI and CyI only gave the desired products in less than 15% yield (detected by crude ^1H -NMR). Moreover, primary alkyl halides were also examined under the reaction conditions, while only gave negative results.

Scheme 1. Cobalt-catalyzed alkylarylation of alkenes. Reaction conditions: Alk-I (0.50 mmol, 2.0 equiv), alkene (0.25 mmol, 1.0 equiv), arylzinc pivalates (0.50 mmol, 2.0 equiv), CoBr_2 (10 mol %), *dppbz* (10 mol %), MeCN (1.5 mL), 35 °C, 16 h.

In addition, we have also repeated the reactions for the difluoroalkylarylation of unactivated alkenes through cobalt catalysis. Please also see the results below (Scheme 2). Furthermore, We have updated the spectra of the compounds 14, 40, 53, 54, 55, 56, 57, 58, 59, 60, 61, 62, 63 and improve the quality of spectra in the revised supporting information.

Scheme 2. Cobalt-catalyzed difluoroalkylarylation of unactivated alkenes.

Reviewer #3 (Remarks to the Author):

Comments: The authors have fulfilled the referees' requests. The paper can be published in Nat. Commun..

Response: We thank this referee's kind comments.

Attached you will find a revised manuscript and the supporting information.

Yours sincerely,

Jie Li & Aiwen Lei

Reviewers' Comments:

Reviewer #2:

Remarks to the Author:

The authors have now addressed my concerns satisfactorily and the manuscript can now be published in Nat Commun in its current form.